# Revealing Spermatogenesis in Smooth-Hound Sharks *Mustelus mustelus*: Insights into the Morphological and Macromolecular Composition of Spermatogenic Cells

**DOI:** 10.3390/ijms25116230

**Published:** 2024-06-05

**Authors:** Giulia Chemello, Lorenzo Jacopo De Santis, Erica Trotta, Matteo Zarantoniello, Chiara Santoni, Francesca Maradonna, Ike Olivotto, Elisabetta Giorgini, Giorgia Gioacchini

**Affiliations:** 1Dipartimento di Scienze della Vita e dell’Ambiente, Università Politecnica delle Marche, 60131 Ancona, Italy; g.chemello@univpm.it (G.C.); e.trotta@pm.univpm.it (E.T.); m.zarantoniello@univpm.it (M.Z.); c.santoni@pm.univpm.it (C.S.); f.maradonna@univpm.it (F.M.); i.olivotto@univpm.it (I.O.); e.giorgini@univpm.it (E.G.); 2Istituto Nazionale Biostrutture e Biosistemi, Consorzio Interuniversitario (INBB), 00136 Rome, Italy; 3National Inter-University Consortium for Marine Science, CoNISMa, 00196 Rome, Italy; l.de.santis@scienze.uniroma2.it

**Keywords:** testis, elasmobranchs, reproduction, Sertoli cell, spermatozoa, smooth-hound shark

## Abstract

Elasmobranchs have an ancestral reproductive system, which offers insights into vertebrate reproductive evolution. Despite their unchanged design over 400 million years, they evolved complex mechanisms ensuring reproductive success. However, human activities induced a significant decline in elasmobranch populations worldwide. In the Mediterranean basin, the smooth-hound shark (*Mustelus mustelus*) is one of the species that are considered vulnerable to human activities. Conservation efforts necessitate a thorough understanding of its reproductive strategy. This study focused on mature male specimens of smooth-hound sharks that were captured in the Adriatic area and successively analyzed to provide, for the first time, a histologically detailed description of testicular development in the species. Seven phases of the spermatogenesis process were identified, along with the macromolecular characterization of cells obtained using Fourier-transform infrared imaging. Histological analysis showed structural and cellular features similar to those documented in the spermatocysts of other elasmobranchs. The examination of the evolution and migration of both germinative and Sertoli cells at each phase revealed their close connection. Furthermore, different expression levels of lipids, proteins, and phosphates (DNA) at each spermatogenesis stage were observed. This research provided new information on spermatogenesis in the common smooth-hound shark, which is crucial for conservation efforts against population decline and anthropogenic pressures.

## 1. Introduction

Elasmobranchs, due to their phylogenetic origin and peculiar reproductive strategies, represent suitable model organisms to investigate the origin, evolution, and basic mechanisms of vertebrates’ reproductive systems.

Generally, fish testes are characterized by two main features that diverge from those of higher vertebrates. Firstly, in fish, Sertoli cells form cysts within spermatogenic tubules, enclosing a single group of germ cells that develop simultaneously. Secondly, Sertoli cells proliferate even in adult fish. The resulting testis structure is characterized by tubules containing cysts at various stages of spermatogenesis [1]. Throughout evolution, teleosts and chondrichthyans have specialized in different reproductive strategies, driven by distinct anatomical and physiological features. Unlike the vast majority of marine teleosts, elasmobranchs exhibit internal fertilization and, most of them, gestational modes that produce a small number of fully developed sharks [2]. Elasmobranchs are currently experiencing a significant and severe decline worldwide under the pressure of human activities [3]. In particular, sharks and rays are subject to direct exploitation as a food source for human consumption in a growing market [4]. Moreover, they are significantly impacted by various fishing techniques, due to which they represent a substantial portion of the bycatch [5]. Globally, it has been estimated that approximately 25% of chondrichthyans are under threat [6].

In the Mediterranean Sea, at least half of the native species of sharks, rays, and chimeras face the risk of extinction [7]. Among them, the smooth-hound shark, *Mustelus mustelus*, along with other members of the same genus, has faced a notable population decline in the Mediterranean Sea as well, related to intensive exploitation by trawlers and small-scale vessels [4,6,8,9]. This species belongs to the Triakidae family and is widespread along the Mediterranean area, including the Adriatic Sea [10]. It presents an annual reproductive cycle during which testis size varies consistently. Similarly to other species of the genus *Mustelus*, different phases of the reproductive cycle may diverge by 1 or 2 months across different regions. Generally, the mating season occurs in the early summer, between May and the beginning of June, and the successive parturition takes place after 3–6 months [11]. For some time now, it has been classified as a vulnerable species in the Mediterranean area according to the IUCN’s Red List of marine species [12]. The northern Adriatic Sea, in particular, is considered an overexploited sub-basin characterized by important anthropogenic pressures and significant environmental changes that threaten elasmobranch survival. In this area, the decline of approximately 80% in elasmobranch landings over the past 68 years testifies to the ongoing reduction in elasmobranch biomass [13,14].

In conservation efforts, it is crucial to thoroughly understand this species’ biology, especially its reproduction. Spermatogenesis is poorly described in the literature among the elasmobranch species, despite their unique testicular organization, making them suitable models for investigating the role and the interactions between Sertoli cells and the germ cells. However, most research into the reproduction of elasmobranchs focuses on females [15,16,17,18], while data on the spermatogenesis of the genus *Mustelus* are scarce and are completely absent for *Mustelus mustelus* [11,19].

Male elasmobranchs possess a distinctive testicular system, comprising testes, genital ducts, urogenital papilla, siphon sacs, and claspers. Testes are paired, symmetrical organs located in the body cavity. Their internal structure is arranged in lobules separated by connective tissue. Within each lobule, multiple spermatocysts, which serve as the functional units of the testis, contain numerous germ cells at the same stage of development [2]. Depending on the location of the germinative area within the lobules, testes can be classified as either radial or diametric, with germinative cells localized either in the center or at one edge of the lobule, respectively. The smooth-hound shark displays a diametric testis development pattern in which spermatogenesis initiates from one wall, generating spermatocysts within the germinal zone, and extends across the diameter of the testis to the opposite wall, where efferent ducts collect the spermatozoa [2]. Each spermatocyst comprises an isogeneic germ cell clone along with a second clonal population of Sertoli cells, with their mitotic divisions being stage-dependent [20]. While the testicular structure and development have been well documented in several shark species with diametric testes, no information is still available regarding the distinctive characteristics of spermatogenesis specific to *M. mustelus*. The present study aimed to present a histological description of the seven main phases that characterize the testicular development in the smooth-hound shark, from spermatogonia to mature spermatozoa. Additionally, for the first time, Fourier-transform infrared imaging (FTIRI) spectroscopy was exploited to evaluate the macromolecular composition of spermatic cysts in sharks during the wave of development. This analysis implemented the current knowledge of *Mustelus*’s spermatogenesis, providing information on the topographical distribution of basic molecules such as proteins, lipids, and phosphates. The correct synthesis and distribution of proteins and lipids during spermatogenesis are indeed pivotal factors influencing the final quality of sperm and, indirectly, the fertilization success from the male perspective [1,21].

## 2. Results and Discussion

Traditional methods to determine sexual maturity and reproductive periods in elasmobranchs, based on macroscopic variations in testicular components, are limited by measurement inaccuracies and varying threshold values among research groups [22,23,24,25]. Therefore, a comprehensive histological and macromolecular analysis of testis species-specific morphology and functional arrangement is needed to unequivocally identify gonadal phases and characterize cell types.

Recently, the nano LC-ESI-MS/MS technique has been used to evidence the proteome differences among four testicular zones corresponding to spermatocysts containing different germinal cell stages of Small-Spotted Catshark (*Scyliorhinus canicula*) [26]. Also, apoptotic events involving both germinal and Sertoli cells, observed during spermatogenesis in some elasmobranch species, may offer valuable information to characterize distinct spermatogenic mechanisms given their species-specific nature [27]. The genus *Mustelus*, similarly to all the members of the Carcharhinidae family, presents a diametric testis, in which spermatogenesis occurs diametrically from the germinal zone (GZ), located on one wall of the testis, to the opposite wall where the efferent ductulus collects the spermatozoa [28,29,30,31]. In this type of testis, the direction of spermatogenesis is clearly distinguishable, with different areas characterized by cysts at the same developmental phase (Figure 1). In mature males, spermatogenesis is characterized by the proliferation and differentiation of both germinative cells and Sertoli cells and occurs through seven different phases, as already reported for other species of the *Mustelus* genus [28,29,30,31]. The count of spermatogenesis phases may vary based on the subjective discretion of different authors. For instance, Park et al. (2013) [32] identified eight spermatogenesis phases for *Mustelus manazo*, designating the degenerative spermatocyst stage as a distinct phase itself.

### 2.1. First Stage: Differentiation of Germ Cells to Spermatogonia

The beginning of spermatogenesis occurs in the germinative zone of this species, as previously described in the dusky smooth-hound shark (*Mustelus canis*) by Conrath and Musick (2002) [31]. Large spermatogonia are loosely organized and can be found isolated, paired, or arranged in clusters within the germinative zone. They could be surrounded by elongated somatic cells, which correspond to Sertoli cell precursors, although no cyst structure is yet visible (Figure 2). The simultaneous presence of different spermatogonia conformations is also evidenced by the IR maps (Figure 3A–D), where high absorbance values of proteins, lipids, and phosphates (DNA) discriminate between single, paired, and grouped large spermatogonia (indicated respectively by white, yellow, and blue arrows).

Moving from the germinative area, spermatogonia and Sertoli cells undergo a series of coordinated mitotic divisions, which are accompanied by the growth and differentiation of the two cell types. The recruitment of spermatogonia and Sertoli cells for the formation of the cyst appeared uniform across all elasmobranch species and differs from the mechanism observed in amniote vertebrates, where Sertoli cell proliferation stops at puberty, reaching a fixed number of “immortal” Sertoli cells [1,33,34].

### 2.2. Second Stage: Early Cyst Proliferation

A newly formed cyst presents a layer of arranged Sertoli cells distributed around the central lumen and, in the beginning, one or two layers of spermatogonia cells (Figure 4 and Figure 5A,B). Spermatogonia proliferate through incomplete cytokinesis; therefore, they are connected through intercellular bridges to form a germ cell syncytium (Figure 5C). The connection between germinative cells has been suggested to synchronize cellular activity in both teleosts and elasmobranchs [1,35]. Mitotic divisions occur until spermatogonia differentiate into primary spermatocytes, resulting in the number of spermatogonia increasing through several incomplete mitotic events. In certain elasmobranch species like *M. mustelus*, the increase in spermatogonia numbers results in six/seven-layered germ cells before differentiation into primary spermatocytes (Figure 6A,B).

### 2.3. Third Stage: Primary Spermatocytes

Primary spermatocytes are frequently found in testis sections, reflecting the long duration of the first meiotic prophase [1]. The differentiation of primary spermatocytes (Figure 7A) is similar among elasmobranchs and involves the migration of Sertoli cell nuclei to the periphery of the spermatocyst, a position maintained throughout subsequent stages. Concurrently, primary spermatocytes undergo the first meiotic division (prophase, metaphase, telophase), during which large nuclei with distinct and elongated chromosomes are visible (Figure 7B) [36]. Challenges in histological analysis restrain the clear identification of meiotic events. Fortunately, infrared (IR) mapping emerges as a valuable tool for highlighting the mitotic spindle, a definitive indicator of ongoing meiotic events (Figure 8A–D). Another distinguishing feature, marking the transition to the primary spermatocyte stage, is the increase in size and number of both primary spermatocytes and Sertoli cells. This corresponds to an enlargement of the cyst diameter, as previously observed in the Brazilian sharpnose shark (*Rhizoprionodon lalandii*) and the blue shark (*Prionace glauca*) [29].

### 2.4. Fourth Stage: Secondary Spermatocytes

Unlike the previous stage, secondary spermatocytes (Figure 7C) are present for only a brief interval between the first and second meiotic divisions, as evidenced by the relatively scarce number of cysts observed at this stage compared to those immediately preceding and following it. Generally, secondary spermatocytes can be differentiated from primary spermatocytes by their comparatively smaller nucleolar size and dense nucleus (Figure 7C,D). These traits are commonly described in all fish, and no peculiar features have been identified between elasmobranch and teleost species [1]. Towards the conclusion of this stage, Sertoli cells become positioned along the periphery of the cyst, as clearly evidenced by the IR maps in Figure 9A–D.

### 2.5. Fifth Stage: Spermatid Types

Three discernible sub-phases, with distinct nuclear morphology, characterized spermatid differentiation (Figure 10A). The same subsequence of the spermatid type was already observed in different elasmobranch species [35,37,38]. The major transformations undergone by spermatids encompass acrosome formation, chromatin compaction, flagella development, cytoplasm reorganization with significant reduction, and the elimination of organelles with the exception of mitochondria. The first sub-phase is characterized by spermatids, which exhibit a round shape and begin to change their orientation towards the cyst border. They present nuclei with dense chromatin and short, distinguishable flagella (Figure 10B). The migration of spermatids towards the periphery of the cyst unequivocally distinguishes this stage from the previous one. The localization of proteins, lipids, and phosphate groups (related to DNA structure) enables us to distinguish spermatids from Sertoli cells, principally based on the distribution of phosphate groups. This differentiation is evident in the high condensation rate of round-shaped spermatids (depicted in pink/white) compared to the lower rate observed in the nuclear area of Sertoli cells (depicted in green) (Figure 11D). In the second sub-phase, spermatid heads elongate with an elliptical shape, and the flagellum (tail) appears easily recognizable (Figure 10C). In the last sub-phase, spermatids organize into bundles, aligning towards the periphery of the cyst (Figure 10D). At this stage, the emergence of the helical head, a characteristic trait of chondrichthyan spermatozoa, becomes evident, together with its connection with Sertoli cells (Figure 12A,B). During this phase of spermatogenesis, this connection has been observed to enable Sertoli cells to effectively remove and recycle residual bodies containing excess cytoplasmic material, which is expelled during the transformation of spermatids into flagellated cells [1].

### 2.6. Sixth Stage: “Tree-Frond-Shaped” Immature Spermatozoa

Immature spermatozoa within each cluster become more compact and align closer to the basal lamina (Figure 13A and Figure 14A). A high compaction degree of the nuclei is clearly highlighted by the IR maps (Figure 14D). The heads of the immature spermatozoa resemble a tree-frond shape and are oriented in their final position, which will persist until the formation of mature sperm. At this stage, the tails of immature spermatozoa are clearly distinguishable, projecting towards the lumen, and acquire the characteristic orientation of a large spiral (Figure 13A). Proteins and lipids are the prominent components of the tails (Figure 14B,C).

At this stage, Sertoli cells are also located along the periphery of the cyst; their number is stabilized compared to the higher proliferation rate observed in cysts preceding the meiotic stage (spermatogonia and spermatocytes I) [29]. Generally, Sertoli cells’ cytoplasm appeared rich in lipids and proteins (red area indicated by yellow arrows in Figure 14B,C), similar to other elasmobranch species previously studied. This macromolecular pattern has been suggested to reflect the presence of a high number of mitochondria and lipid droplets and an abundant smooth endoplasmic reticulum related to the steroidogenesis activity of the Sertoli cells [20,39].

### 2.7. Seventh Stage: Mature Spermatozoa

In this phase, a major degree of compaction is observed in the grouped heads of spermatozoa that leads to transitioning from a loosely packed arrangement of the tree-frond-shaped immature spermatozoa into a high-density arrowhead shape (Figure 13B,C and Figure 15A,D). These tightly packed formations arrange themselves in a spiral pattern along the borders of the cysts. As for the tails, they maintain their spiral orientation, exhibiting a more compact structure. Furthermore, the macromolecular composition of the tails showed alterations at this stage. In fact, IR maps still revealed a high concentration of lipids, accompanied by an increased concentration of proteins in the central area of the cyst (Figure 15B,C). At this point, mature spermatozoa can be released through the collecting ductulus. Based on what has been observed in mammals, the maturation of spermatozoa should occur in the epididymis, where they transit, thanks to the presence of ciliated epithelial cells in the lumen of branched, stem, and collecting ducts and the contraction of myoid cells surrounding the intratesticular ducts [40]. Unfortunately, the epididymis in elasmobranchs has been poorly studied despite its size and anatomical similarity to that of mammals. Studies in the Port Jackson Shark (*Heterodontus portusjacksoni*) and in the Clearnose skate (*Raja eglanteria*) have shown that epididymis effectively participates in luminal fluid modifications, including sodium resorption, and that spermatozoa, immotile in the proximal segment, acquired their motility during their transit in the terminal segment of epididymis [40]. After the release of spermatozoa, what remains is an empty cyst, which contains basal cytoplasm characterized by a moderate concentration of proteins and lipids (Figure 16B,C) and the nuclei of Sertoli cells (Figure 13D and Figure 16D). These remaining cellular residuals will degenerate and be reabsorbed, probably through the hemophagocytosis process by leukocytes, which may be recruited from the epigonal organ [32]. This process involves apoptotic events that affect the nuclei of Sertoli cells, resulting in their fragmentation and the subsequent manifestation of uniform secondary necrosis [33,41]. Differently, it has been observed that the cytoplasm of the Sertoli cells, named cytoplast, together with other cellular debris is normally released from the cysts together with mature spermatozoa [42,43].

## 3. Materials and Methods

### 3.1. Sample Collection

Twenty *Mustelus mustelus* male specimens were collected during May and June of 2019 and 2021 in collaboration with the local fishing fleet of Ancona by trawling within the FAO-Geographical Sub-Area 17 of the Northern–Central Adriatic Sea. Species identification was performed following the merged guidelines based on specific morphological features described by Marino et al. (2015) and Golani et al. (2007) [19,44]. Mature males were identified based on the total length and macroscopic analysis of their testes. According to the L50 observed in a previous study performed in the North-Eastern Mediterranean [45], males with a total length > 92 cm were considered to be examined through histological analysis of the testis to confirm the sexual maturity of the males selected. Only adult specimens were considered to describe the spermatogenesis phases.

### 3.2. Histological Analysis

From each male specimen, the central portion of each testis was sampled and stored in a formaldehyde/glutaraldehyde solution (NaH_2_PO_4_–H_2_O + NaOH + Formaldehyde (36.5%) + Glutaraldehyde (25%) + H_2_O) and stored at −4 °C for histological and FTIR analyses. Samples were successively processed according to Chemello et al. (2023) [46], as briefly reported below. Samples stored in the fixative solution were embedded in paraffin blocks, and successively, sections with a thickness of 5 µm were obtained from each sample using a microtome (Leica, RM2125 RTS, Nussloch, Germany) and positioned on glass slides. Finally, sections were stained with hematoxylin and eosin Y stain (Merck KGaA, Darmstadt, Germany) and then observed using an optical microscope (Zeiss Axio Imager.A2, Oberkochen, Germany). Images at different magnifications were acquired with a combined color digital camera, the Axiocam 503 (Zeiss, Oberkochen, Germany).

### 3.3. Fourier-Transform InfraRed Imaging (FTIRI) Analysis

The same samples used for histological analysis were also submitted to FTIRI. To this purpose, from each paraffin-embedded block, two 5 µm thick sections were cut using a microtome and deposited onto CaF2 optical windows. The analysis was performed using a spectrometer composed of an Invenio R interferometer coupled with a Hyperion 3000 Vis-IR microscope (Bruker Optics, Ettlingen, Germany). The visible image of each section was obtained with a 15X condenser/objective. On the selected areas, corresponding to spermatocysts at different stages, IR images were acquired in transmission mode in the 4000–900 cm^−1^ range (spectral resolution 4 cm^−1^) by using a liquid nitrogen-cooled bidimensional focal plane array (FPA) detector. This array detector, suitable for the morpho-chemical analysis of non-homogeneous biological samples, lets generate IR images composed of blocks of 4096 pixel/spectra, simultaneously acquired on an area of 164 × 164 μm^2^; each pixel/spectrum represented an area of 2.56 × 2.56 μm^2^ and was the result of 256 scans. The size of the mapped areas was chosen based on the cysts’ dimensions. Background spectra were obtained from clean regions of the CaF2 optical windows.

The atmospheric compensation routine was applied to correct raw IR images, to remove the contribution of atmospheric carbon dioxide and water vapor, and then vector normalized in the 4000–900 cm^−1^ spectral range to avoid artifacts deriving from thickness differences (OPUS 7.1 software, Bruker Optics, Ettlingen, Germany). These pre-processed IR images were integrated under the following spectral regions to obtain false color maps representing the topographical distribution and relative amount of the most relevant biochemical components: 2995–2825 cm^−1^ representative of lipids; 1718–1481 cm^−1^ representative of proteins; and 1274–1181 cm^−1^ representative of phosphate groups in nucleic acids. To build false color maps, an arbitrary color scale was used, with white indicating areas with the highest absorbance values and blue indicating areas with the lowest ones.

## 4. Conclusions

This study provided a comprehensive overview of the morphological, structural, and molecular configuration of all stages characterizing the sperm development of the smooth-hound shark, highlighting the unique characteristics that distinguish this species from other elasmobranchs. The information provided by this study is essential to better understanding the morphological and biochemical changes during the spermatogenesis process of the smooth-hound shark. The exhaustive characterization of this process could represent a valid tool for the study of possible spermatogenesis impairment due to environmental changes. This aspect is crucial in terms of conservation since the effects of overexploitation could be worsened by the more frequent environmental alterations that characterize the present era. Moreover, infrared imaging represents an innovative tool to investigate cellular mechanisms that underlie essential processes for the survival and growth of a species, not only reproduction.

## Figures and Tables

**Figure 1 ijms-25-06230-f001:**
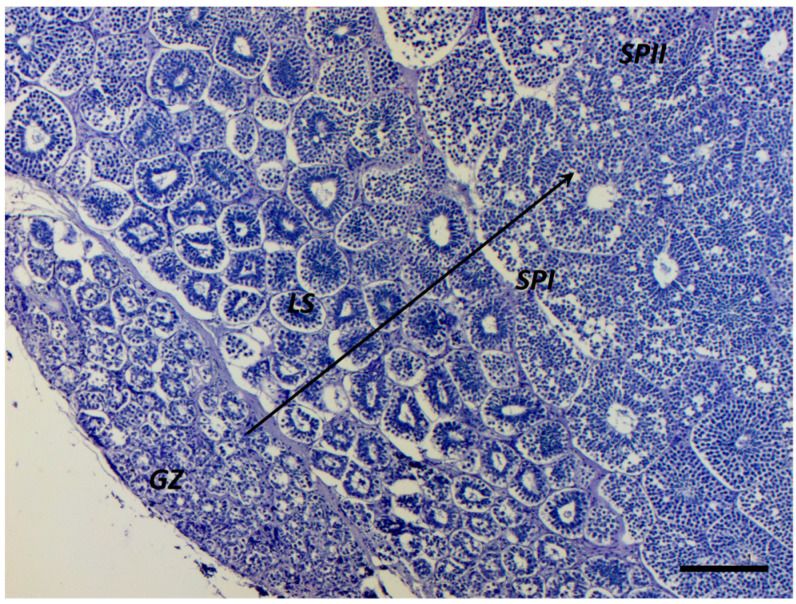
Panoramic image of the *Mustelus mustelus* diametric testis. The black arrow indicates the progression of germinative cell maturation, starting from the germinative zone (GZ), followed by cysts containing spermatogonia (LS), primary spermatocytes (SPI), and secondary spermatocytes (SPII). Scale bar: 200 µm.

**Figure 2 ijms-25-06230-f002:**
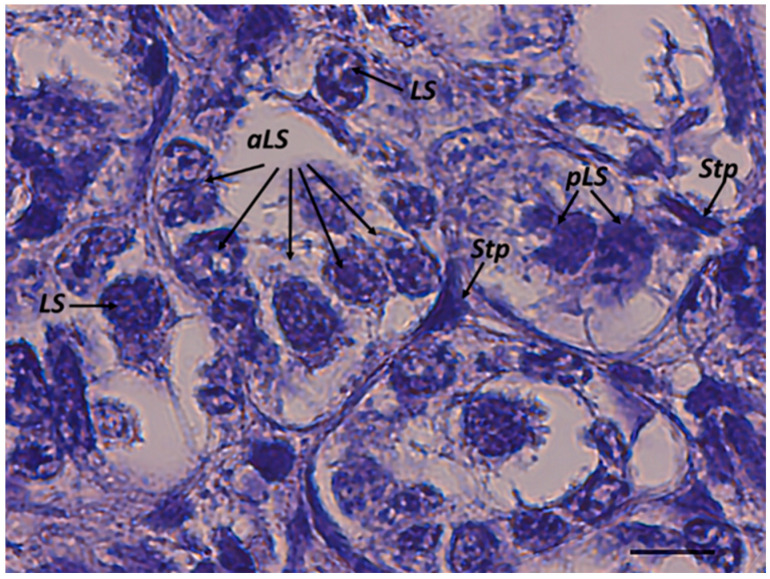
Large spermatogonia rearranging with Sertoli cell precursors in the germinative zone. Stp, Sertoli cell precursor; LS, large single spermatogonia; pLS, paired large spermatogonia; aLS, arranged large spermatogonia. Scale bar: 10 µm.

**Figure 3 ijms-25-06230-f003:**
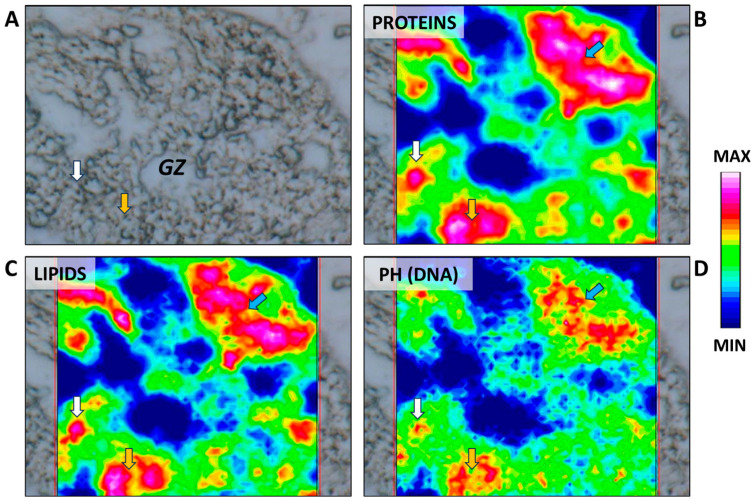
(**A**) Microphotograph representative of the germinative zone (GZ). False-color IR maps show the topographical distribution of (**B**) proteins, (**C**) lipids, and (**D**) phosphate groups (PH). White arrows indicate single spermatogonia; yellow arrows indicate paired spermatogonia; and blue arrows indicate grouped spermatogonia. A color scale was used to highlight different absorbance values: blue indicates areas with the lowest ones, while white indicates those with the highest ones.

**Figure 4 ijms-25-06230-f004:**
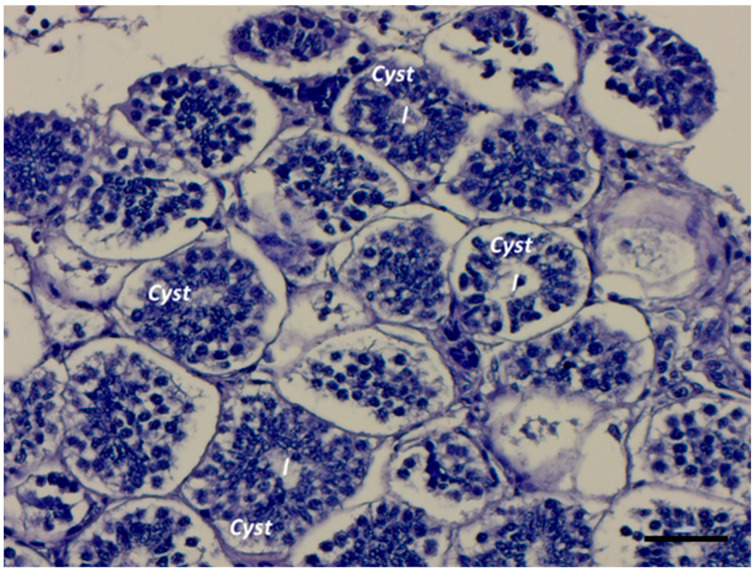
Early cyst formation through aggregation of large spermatogonia and Sertoli cells arranged around the central lumen of the newly formed cyst; l, the cyst’s lumen. Scale bar: 50 µm.

**Figure 5 ijms-25-06230-f005:**
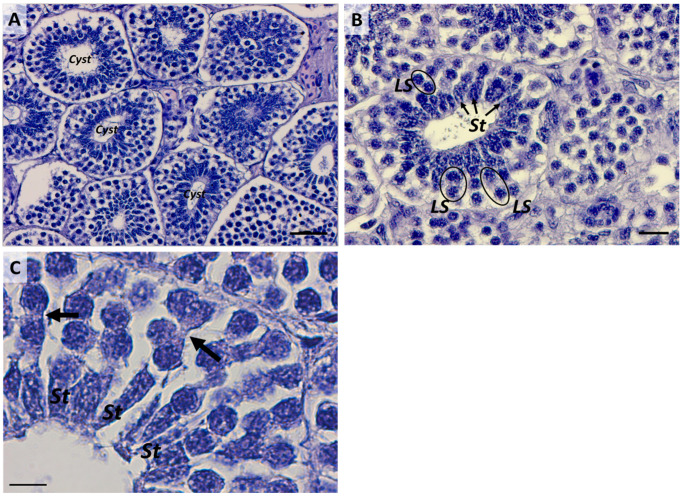
(**A**) Early spermatocyst phase. Scale bar: 50 µm. (**B**), Structural details of an early spermatocyst characterized by large spermatogonia (LS) (black circles) arranged in 2 layers connected to the single layer of Sertoli cells (St) (black arrows) distributed around the cyst’s lumen. Scale bar: 20 µm. (**C**) Cell syncytium (black arrows) in an early spermatocyte. Scale bar: 10 µm.

**Figure 6 ijms-25-06230-f006:**
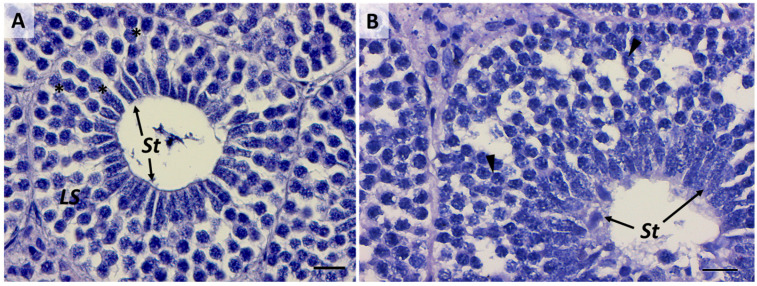
(**A**) Cyst in progression from stage 2 to stage 3 with four spermatogonia layers (black asterisks) and (**B**) five and more spermatogonia layers (black arrowheads). St, Sertoli cell; LS, large spermatogonia. Scale bars: 20 µm.

**Figure 7 ijms-25-06230-f007:**
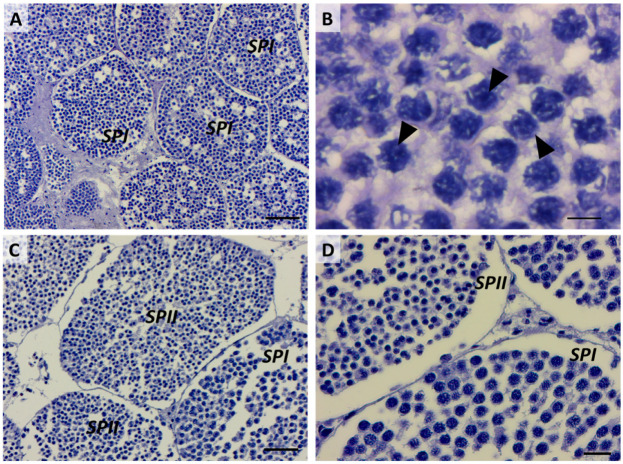
(**A**) Primary spermatocytes. Scale bar: 100 µm. (**B**) Detailed image of the nuclei (black arrowheads) of primary spermatocytes. Scale bar: 10 µm. (**C**) Comparison between primary and secondary spermatocytes. Scale bar: 50 µm. (**D**) Details of primary and secondary spermatocytes’ nuclei. Scale bar: 20 µm. SPI, primary spermatocytes; SPII, secondary spermatocytes.

**Figure 8 ijms-25-06230-f008:**
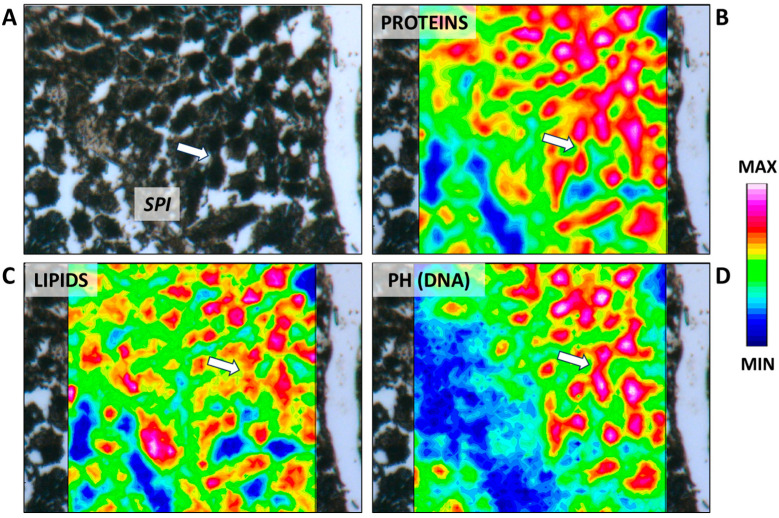
(**A**) Microphotograph representative of the primary spermatocyte stage (SPI). False-color IR maps show the topographical distribution of (**B**) proteins, (**C**) lipids, and (**D**) phosphate groups (PH). The white arrow indicates the ongoing meiotic division of a primary spermatocyte. A color scale was used to highlight different absorbance values: blue indicates areas with the lowest ones, while white indicates those with the highest ones.

**Figure 9 ijms-25-06230-f009:**
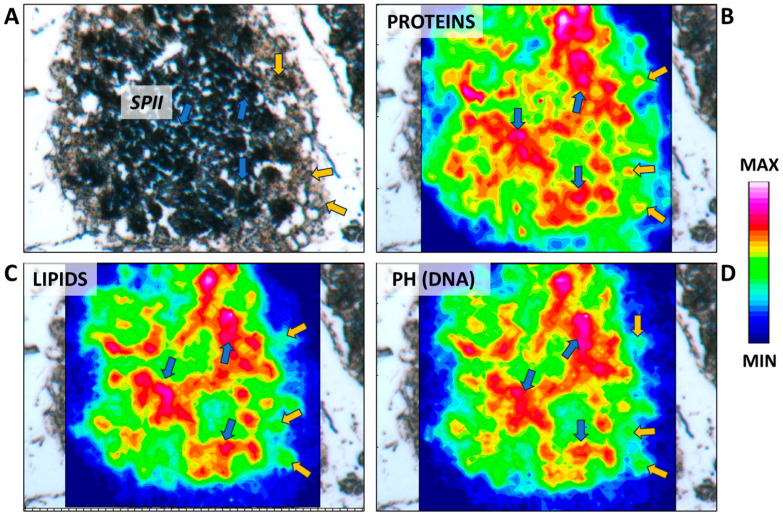
(**A**) Microphotograph representative of the secondary spermatocyte stage (SPII). False-color IR maps show the topographical distribution of (**B**) proteins, (**C**) lipids, and (**D**) phosphate groups (PH). Blue arrows indicate secondary spermatocytes, and yellow arrows indicate Sertoli cells. A color scale was used to highlight different absorbance values: blue indicates areas with the lowest ones, while white indicates those with the highest ones.

**Figure 10 ijms-25-06230-f010:**
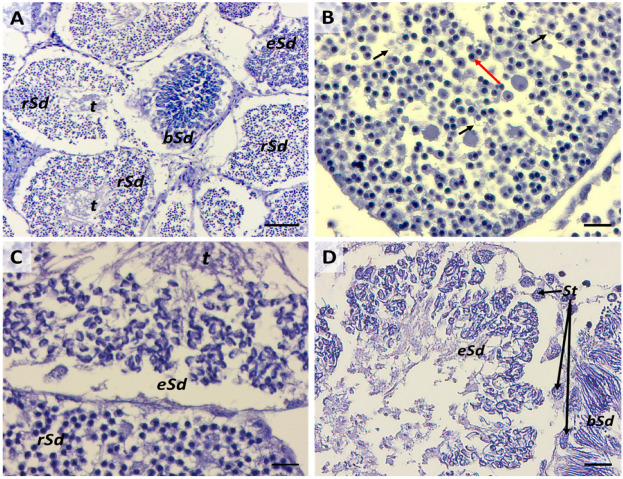
(**A**) Cysts at different spermatid sub-phases. Scale bar: 100 µm. (**B**) Round-shape spermatids with visible condensed chromatin (red arrow) and preliminary flagella (black arrows). Scale bar: 20 µm. (**C**) Comparison between cysts with round-shape and elliptic-shape spermatids. Scale bar: 20 µm. (**D**) Cysts with elliptic-shape and bundle-shape spermatids. Scale bars: 20 µm; t, spermatids’ tails; rSd, round-shape spermatid; eSd, elliptic-shape spermatid; bSd, bundle-shape; St, Sertoli cell.

**Figure 11 ijms-25-06230-f011:**
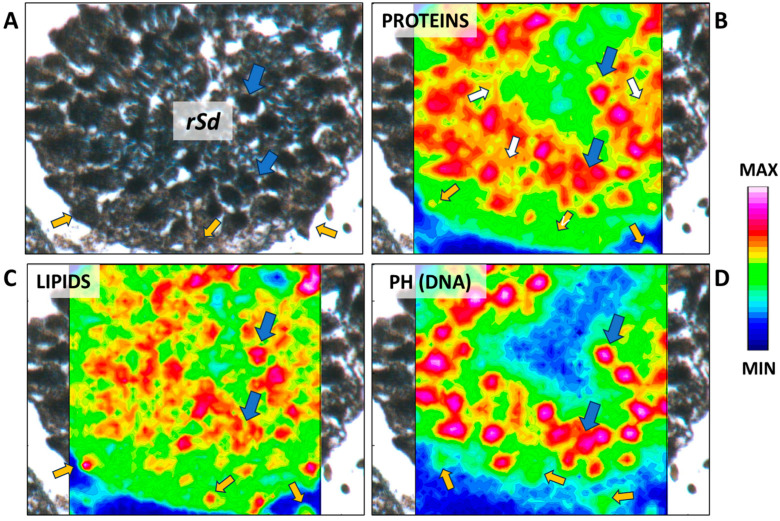
(**A**) Microphotograph representative of the round-shape spermatids (rSd). False-color IR maps show the topographical distribution of (**B**) proteins, (**C**) lipids, and (**D**) phosphate groups (PH). Blue arrows indicate round-shape spermatids, yellow arrows indicate Sertoli cells, and white arrows indicate flagella precursors. A color scale was used to highlight different absorbance values: blue indicates areas with the lowest ones, while white indicates those with the highest ones.

**Figure 12 ijms-25-06230-f012:**
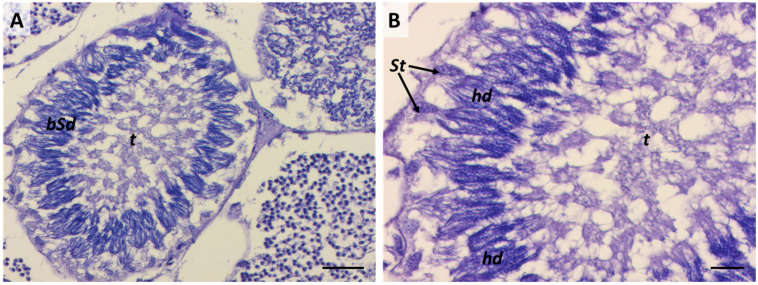
(**A**) Cyst containing bundle-head spermatid sub-phase; scale bars: 50 µm. (**B**) Visible connection between Sertoli cells and the heads of bundle-shape spermatids (hd); t, spermatids’ tails. Scale bars: 20 µm.

**Figure 13 ijms-25-06230-f013:**
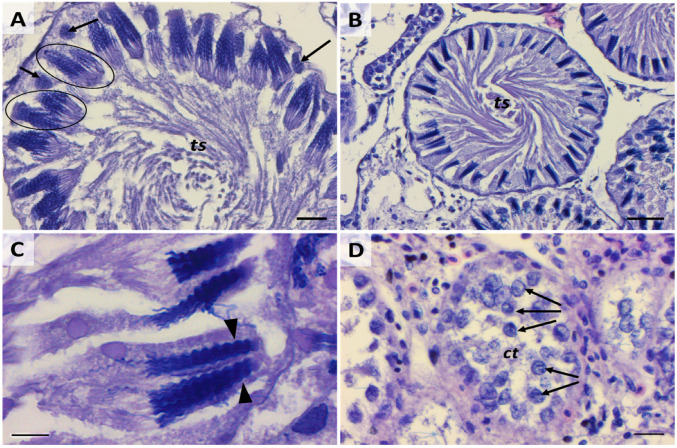
(**A**) Cyst containing immature spermatozoa with the typical tree-frond head shape (black circles) and Sertoli cells (black arrows). Scale bar: 20 µm. (**B**) Cyst containing mature spermatozoa. Scale bar: 50 µm. (**C**) Particular arrow-shape heads of mature spermatozoa (black arrowheads). Scale bar: 10 µm. (**D**) Empty cysts containing Sertoli cells (black arrows). Scale bar: 20 µm; ts, spiral of tails; ct, cytoplasm.

**Figure 14 ijms-25-06230-f014:**
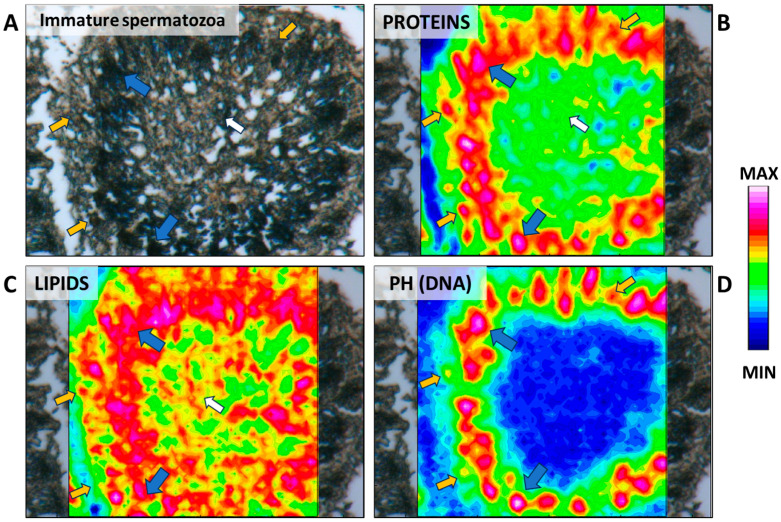
(**A**) Microphotograph representative of an immature spermatozoa. False-color IR maps show the topographical distribution of (**B**) proteins, (**C**) lipids, and (**D**) phosphate groups (PH). Blue arrows indicate the immature spermatozoa nuclei; yellow arrows indicate Sertoli cells; and white arrows indicate the tails of immature spermatozoa. A color scale was used to highlight different absorbance values: blue indicates areas with the lowest ones, while white indicates those with the highest ones.

**Figure 15 ijms-25-06230-f015:**
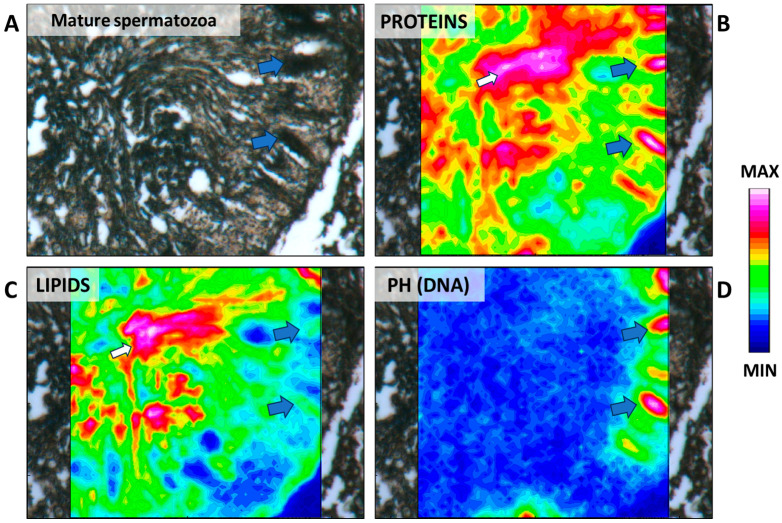
(**A**) Microphotograph representative of a mature spermatozoa. False-color IR maps show the topographical distribution of (**B**) proteins, (**C**) lipids, and (**D**) phosphate groups (PH). Blue arrows indicate the mature spermatozoa nuclei; and white arrows indicate the tails of mature spermatozoa. A color scale was used to highlight different absorbance values: blue indicates areas with the lowest ones, while white indicates those with the highest ones.

**Figure 16 ijms-25-06230-f016:**
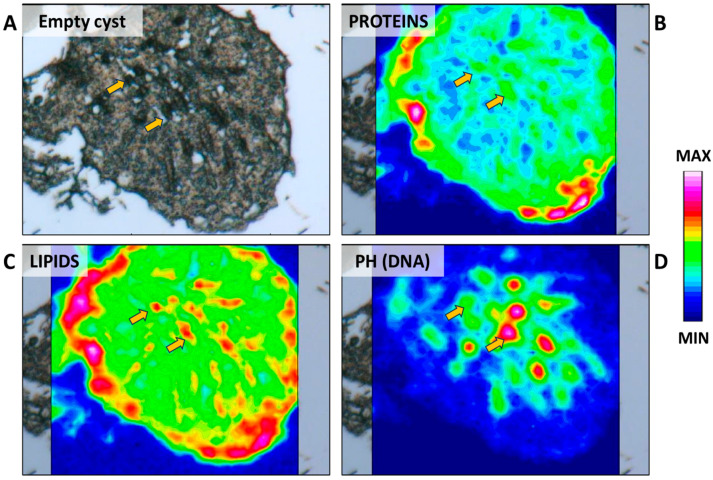
(**A**) Microphotograph representative of an empty cyst. False-color IR maps show the topographical distribution of (**B**) proteins, (**C**) lipids, and (**D**) phosphate groups (PH). Yellow arrows indicate Sertoli cells. A color scale was used to highlight different absorbance values: blue indicates areas with the lowest ones, while white indicates those with the highest ones.

## Data Availability

The raw data supporting the conclusions of this article will be made available by the authors on request.

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
