# Peer review of "Revealing Spermatogenesis in Smooth-Hound Sharks Mustelus mustelus: Insights into the Morphological and Macromolecular Composition of Spermatogenic Cells"

_ijms, 2024, doi:10.3390/ijms25116230_

Round 1

Reviewer 1 Report

Comments and Suggestions for Authors

In this current manuscript, Chemello and co-authors have investigated the spermatogenic progression in adult smooth-hound shark Mustelus mustelus. The study mainly focuses on the histological details (along with macromolecular composition) of seven different phases of spermatocyst development. Despite being a novel study (with limited information available in the literature) some critical points are to be addressed /added prior to its final acceptance.

1. Line-47-51, the rationale of using smooth-hound shark M. mustelus as model elasmobranch is not welljustified. In addition to Reference 5 and 6, more strong and relevant references canbe cited.

2. Line 56-70, the key features of spermatogenic development found in elasmobranchs are to be discussed with more clarity.

3. Please mentiontheuniquefeaturesoftesticular developmentinfishes.

4. The unique mode of spermatocyst maturationof this species andhowitdiffersfromother elasmobranchs are not appropriatelyhighlighted/ discussed. A distinct comparison with other marine teleost found in the Adriatic Sea can be alsobediscussed.

5. How many individual fish samples were collected?

6.Line 323-325, as all the samples were captured from the natural/ wild habitat, how these samples were examined to be considered as same species category? Kindly add a note on the morphometric and/or molecular basis of the taxonomical identification.

7. Why TEM and SEM imaging were not included in this study? Please justify this point. If possible (as per the availability of old and/or fresh samples) kindly generate/add some new data.

8. The Annual testicular cyclicity has been largely ignored. Kindly add a note on the seasonal variations of testicular development.

9. Line 367-68, not clear. Kindly elaborate the overall outcome of this research.

10. Provide a graphical summary of this research.

Comments on the Quality of English Language

.

Reviewer 2 Report

Comments and Suggestions for Authors

MS is very interesting and useful for the development of science. Testis development and sperm maturation have been described in MS. This is a very important aspect of the reproduction of each species. However, in the case of fish, data on spermatogonia are scarce because many authors focus mainly on females.

However, I lack information as to why the authors examined the distribution of proteins and lipids in the testicles. What role do proteins and proteins play in sperm development? This information should be included at least in the Introduction section.

Reviewer 3 Report

Comments and Suggestions for Authors

The title should contain the common name and change it to "Revealing spermatogenesis of smooth-hound sharks, Mustelus mustelus: insights into morphological and macromolecular composition of spermatogenic cells

This paper would be a better fit for a morphology journal.  

This is a descriptive paper and not testing a hypothesis.  Delineate it near Ln.76.

Figs are too small throughout. 

What is the sample size?  How many males?  Is each image a representative of the average view?  What is the reproductive age of the animals - if they are not all the same, then these data could be flawed.  Where the animals the same total length?  These factors should be clearly described in the methods.

That conservation and anthropogenic impacts are of concern is not well integrated with the intent of the paper which is descriptive.  Which list is used for considering this species as vulnerable?  Specify the exact list.  

Throughout the paper there are run-on paragraphs were a new paragraph is needed, such as Ln. 40, 45, 54,61,92,

Ln. 34 "peculiar reproduction mode" is unclear

Ln. 43.  Citation needed for the market demand.

Comments on the Quality of English Language

OK - slight grammar and tightening needed

Round 2

Reviewer 1 Report

Comments and Suggestions for Authors

The manuscript has been improved after revision, it can be accepted for publication.

Author Response

The Authors would like to thank the reviwer for their last comment.

Reviewer 3 Report

Comments and Suggestions for Authors

Thanks to the authors for carefully addressing comments.

Line 79 on the marked version could be rephrased from "However, the main notions regarding reproductions of elasmobranchs refer to female individuals...." to "However, most research into reproduction of elasmobranchs focuses on females, while studies ....."

Images could be brightened up.

Comments on the Quality of English Language

Slight edits and condensation and more paragraph starts would improve readability.

Author Response

Replies to comments and Suggestions for Authors

- Line 79 on the marked version could be rephrased from "However, the main notions regarding reproductions of elasmobranchs refer to female individuals...." to "However, most research into reproduction of elasmobranchs focuses on females, while studies ....."

Authors’ reply: As suggested by the reviewer the text has been modified (See line 79 of the revised manuscript).

- Images could be brightened up.

Authors’ reply: All the images have been revised and when possible, they have been brightened up avoiding the alteration of their clarity (See the revised manuscript).

 Comments on the Quality of English Language

 - Slight edits and condensation and more paragraph starts would improve readability.

Authors’ reply: As suggested by the reviewer, the entire manuscript has been revised to improve its readability.